# Precise Detection and Visualization of Nanoscale Temporal Confinement in Single-Molecule Tracking Analysis

**DOI:** 10.3390/membranes12070650

**Published:** 2022-06-24

**Authors:** Manon Westra, Harold D. MacGillavry

**Affiliations:** Cell Biology, Neurobiology and Biophysics, Department of Biology, Faculty of Science, Utrecht University, 3584 CH Utrecht, The Netherlands; m.westra@uu.nl

**Keywords:** single-molecule tracking, confinement, plasma membrane

## Abstract

The plasma membrane consists of a diverse mixture of molecules that dynamically assemble into a highly non-random organization. The formation of nanoscale domains in the membrane is of particular interest as these domains underlie critical cellular functions. Single-molecule tracking is a powerful method to detect and quantify molecular motion at high temporal and spatial resolution and has therefore been instrumental in understanding mechanisms that underlie membrane organization. In single-molecule trajectories, regions of temporal confinement can be determined that might reveal interesting biophysical interactions important for domain formation. However, analytical methods for the detection of temporal confinement in single-molecule trajectories depend on a variety of parameters that heavily depend on experimental factors and the influence of these factors on the performance of confinement detection are not well understood. Here, we present elaborate confinement analyses on simulated random walks and trajectories that display transient confined behavior to optimize the parameters for different experimental conditions. Furthermore, we demonstrate a heatmap visualization tool that allows spatial mapping of confinement hotspots relative to subcellular markers. Using these optimized tools, we reliably detected subdiffusive behavior of different membrane components and observed differences in the confinement behavior of two types of glutamate receptors in neurons. This study will help in further understanding the dynamic behavior of the complex membrane and its role in cellular functioning.

## 1. Introduction

The plasma membrane is a highly complex and dynamic environment where a vast variety of transmembrane proteins are embedded in a mixture of over a hundred different types of lipids. Key to understanding membrane organization is determining how components are organized and move within the lateral plane of the membrane [1]. The development of single-molecule tracking techniques has been instrumental in quantifying the diffusion of membrane components in living cells and has provided important new insights into how the dynamic nanoscale organization of membrane components contributes to cellular functions. In contrast to other techniques, single-molecule tracking experiments provide trajectories that describe the motion of individual molecules, rather than the average, ‘ensemble’ behavior of a population of molecules. Careful analysis of single-molecule trajectories can therefore reveal a wealth of information on the dynamic behavior of molecules, the biophysical properties of the cellular environment, and the compartmentalization of molecules. Single-molecule trajectories can be described as directed, random (Brownian), or confined motion. Confinement zones, regions where a molecule remains longer than expected from a Brownian diffusant, are of particular biological interest. Confinement zones could indicate organizational hotspots where proteins undergo transient binding to intracellular scaffold molecules or are trapped by the underlying membrane cytoskeleton. Examples of such organizational hotspots that are relevant for biological processes are lipid domains [2,3], neurotransmitter receptor nanodomains [4,5,6,7], G-protein-coupled receptor hotspots [8] and ion channel nanoclusters [9,10]. There is thus a need for analytical tools that reliably detect, measure, and visualize confinement zones.

Several studies have reported on strategies to analyze and detect confined behavior in single-molecule trajectories [11,12,13,14,15,16,17,18]. In fact, Einstein formulated a theory about Brownian diffusants already in 1905, where he argued that the displacement of a Brownian particle is proportional to the square root of the elapsed time [19]. Confinement is defined as the portion of a trajectory that deviates from what a random walk would look like. Therefore, it is essential to know how long a Brownian diffusant would stay in a certain region. Saxton defined the probability that a molecule will stay in a region by the following equation: $\log\psi =0.2048-2.5117\left(D\times t/R^{2}\right)$ where *D* is the diffusion coefficient, *t* the period of time, and *R* the radius of the region [20]. Simson et al. translated this probability into the confinement index, which is inversely related to the probability that a Brownian molecule will stay in a certain area [11]. When this probability during the trajectory becomes lower, the confinement index will increase, indicating a period of transient confinement within the trajectory. This confinement analysis has been used extensively by many labs to detect temporal confinement in single-molecule trajectories [21,22,23,24,25,26,27,28,29,30,31,32,33]. 

This confinement analysis depends on different manually defined parameters that need to be optimized to reliably detect confinement. These parameters include for instance the window of frames to analyze, threshold for the confinement index, and the minimal time a molecule should be in this state to be considered confined. Determining the optimal values for each of these parameters, however, is not trivial and varying individual parameters can have a large impact on the detection power of the analysis. Therefore, in this study, we set out to understand the influence of individual parameters on the performance of the confinement analysis and tested the robustness of this analysis on simulated random walks and trajectories that display transient confined behavior. This allowed us to optimize parameters and balance the detection of false-negatives and false-positives. Furthermore, we developed a tool to visualize confinement areas in heatmaps that allows spatial mapping of confinement hotspots relative to subcellular markers. To test the performance of this analysis, we applied our analysis on experimental data and reliably detected subdiffusive behavior for a variety of membrane components. Lastly, we found that two neuronal glutamate receptors, mGluR5 (metabotropic glutamate receptor 5) and AMPA-type glutamate receptors, reveal different confinement properties. This study will help in further understanding the dynamic behavior of membrane components and their role in membrane organization.

## 2. Materials and Methods

### 2.1. Simulations

To simulate 2D random walks, we tested two different simulation methods (Appendix A). In the first model (model 1), every consecutive coordinate is drawn from a Gaussian distribution based on the diffusion coefficient and the interval time between the steps: $step=\sqrt{2\times D\times \Delta t}\times \left[x\;y\right]$ where *x* and *y* are normally distributed numbers, generated by the MATLAB function *randn*. In the second model (model 2), every step has the same step size derived from the set diffusion coefficient and interval time, however the angle and thus the direction of the step is random: $step=\sqrt{4\times D\times \Delta t}\times \left[\cos\left(\theta \right)\;\sin\left(\theta \right)\right]$ where *θ* is a random angle between 0 and 2*π,* generated using MATLAB (MathWorks, Natick, MA, USA, R2021B) function *rand* (Appendix A). The MSD curve of model 2 showed less variation in the curve, however the estimated diffusion coefficient is for both models highly similar to the set diffusion coefficient for the simulation. We applied subsampling to the simulated tracks so that every 100 steps the coordinates were saved in the trajectory. After subsampling, the distributions of the diffusion coefficients were almost identical for both models (Appendix A) and for the rest of the simulations, we used model 2.

We also tested two different approaches for simulating transient confined trajectories. The principal idea behind the simulations is that the molecule cannot escape a simulated circle for as long as it is simulated to be confined. The first approach is the ‘reflect’ model, where the molecule bounces back from the simulated, circular confinement zone if the next step would be outside the circle (Appendix A, left panel). The second approach is the ‘stick’ model, where the molecule sticks at the border at the place where it would otherwise escape the circle (Appendix A, right panel). The ‘reflective’ model appeared to be more stable when varying the step sizes in comparison to the ‘stick’ model. When increasing the step size in the simulations, the plateau in the MSD curve reached higher values for the ‘stick’ model, while it remained the same for the ‘reflect’ model (Appendix A). Therefore, we used the reflective model for the rest of the confinement simulations. 

### 2.2. Measure Power of Detection

The power of detection was measured as the percentage of confined displacements, meaning all steps in all trajectories simulated that are in a period of detected confinement, i.e., above threshold *Lc* for period of time longer than *Tc*.

### 2.3. Parameters Used in Simulations

Simulations were performed to generate trajectories of 100 frames and confinement periods of 0, 15, or 50 frames, with diffusion coefficient of 0.05 μm^2^/s and interval time of 0.02 s unless reported otherwise. Basic simulation settings: *D* = 0.05 μm^2^/s, *Nsteps* = 10,000, subsampling = 100, *dt* = 0.02 s, *dt_sim* = 0.0002 s. Basic confinement parameters: *Lcm* = 5.5, *Sm* = 15, *Smin* = 4, *Tc* = 0.2 s, α = 0.5, npointsMSDset = 10, *Rconf* = 0.04 μm (Appendix A).

### 2.4. Animals

All animal experiments were performed in compliance with the guidelines for the welfare of experimental animals issued by the Government of the Netherlands (Wet op de Dierproeven, 1996) and European regulations (Guideline 86/609/EEC). All animal experiments were approved by the Dutch Animal Experiments Review Committee (Dier Experimenten Commissie; DEC), performed in line with the institutional guidelines of Utrecht University.

### 2.5. Primary Neuron Cultures and Transfections

Dissociated hippocampal cultures were prepared from embryonic day 18 (E18) Janvier Wistar rat brains of both genders [34]. Dissociated neurons were plated on Ø18-mm coverslips coated with poly-L-lysine (37.5 µg/mL, Sigma-Aldrich, St. Louis, MO, USA) and laminin (1.25 µg/mL, Roche Diagnostics, Mannheim, Germany) at a density of 100,000 neurons per well. Neurons were grown in Neurobasal medium (NB) supplemented with 1% penicillin/streptomycin (Gibco), 2% B27 (Gibco), and 0.5 mM L-glutamine (Gibco, Grand Island, NY, USA) at 37 °C in 5% CO_2_. From days in vitro (DIV) 1 onward, medium was refreshed weekly by replacing half of the medium with BrainPhys Neuronal Medium (BP, STEMCELL Technologies, Vancouver, Canada) supplemented with 2% NeuroCult SM1 (STEMCELL Technologies, Vancouver, Canada) and 1% penicillin/streptomycin. At DIV14, neurons were transfected with indicated constructs using Lipofectamine 2000 (Invitrogen, Carlsbad, CA, USA). For each well, 1.8 μg DNA was mixed with 3.3 μL Lipofectamine 2000 in 200 μL BP, incubated for 30 min at room temperature (RT). Then, 400 μL conditioned medium was transferred to a new culture plate and replaced by 260 μL BP supplemented with 0.5 mM L-glutamine. The DNA mix was added to the neurons and incubated at 37 °C in 5% CO_2_. After 1 to 2 h, neurons were briefly washed with BP and transferred to the new culture plate with conditioned medium supplemented with an additional 400 μL BP with SM1 and penicillin/streptomycin and kept at 37 °C in 5% CO_2_. All experiments were performed using neurons at DIV18-22. 

### 2.6. DNA Plasmids

All plasmids have been described before in previous studies: Homer1c-mCherry [35], SEP-GluA2 [4], SEP-mGluR5 [35]. The GFP-GPI and mHD-GT46 constructs are kind gifts from Dr. Ewers (Institute for Chemistry and Biochemistry, Free University Berlin, Germany; [36]).

### 2.7. Single-Molecule Tracking

Single-molecule tracking was performed on the Nanoimager S from ONI (Oxford Nanoimaging; ONI, Oxford, UK), equipped with a 100×/NA 1.4 oil immersion objective (Olympus Plan Apo, Hamburg, Germany), an XYZ closed-loop piezo stage, and with 471-, 561-, and 640-nm wavelength excitation lasers. Fluorescence emission was detected using a sCMOS camera (ORCA Flash 4, Hamamatsu, Hamamatsu City, Japan). Stacks of 5000 frames were acquired at 50 or 100 Hz with oblique illumination using the 640-nm laser. NimOS software (version 1.18.3) from ONI was used for localization and drift correction was performed. Neurons were imaged in extracellular imaging buffer containing 120 mM NaCl, 3 mM KCl, 10 mM HEPES, 2 mM CaCl2, 2 mM MgCl_2_, and 10 mM glucose, pH adjusted to 7.35 with NaOH. The GFP/RFP-booster Atto647N (Chromotek, Martinsried, Germany) was added before image acquisition in a concentration of 1:200,000 in extracellular imaging buffer while blocking with 1% BSA. For the tracking of DOPE, we added Atto647N-DOPE (ATTO-TEC, Siegen, Germany) in complex with defatted BSA [37] before image acquisition at a final concentration of 6.5 nM. We used a minimum track length of 30 frames (10- or 20-ms interval) for visualization and quantification. Postsynaptic density (PSD) masks were created from a stack of 30 frames obtained for Homer1c-mCherry using the 561 nm excitation laser as described in [38].

### 2.8. Single-Molecule Tracking Analysis

Using MATLAB (version 2021b), molecules with a localization precision <50 nm were selected for analysis. Tracking was achieved using custom algorithms in MATLAB described previously with a tracking radius of <500 nm [39]. The first three points of the MSD with the addition of the value 0 at MSD(0) were used to fit the slope using a linear fit. Tracks with a negative slope were not used for further analysis. The diffusion coefficient was estimated based on the fit using the formula: MSD=4×D×Δt. Only tracks of at least 30 frames were selected for further analysis. Tracks were classified as immobile when the ratio between the radius of gyration and mean step size ($\frac{\sqrt{\pi /2}\cdot radius\;of\;gyration}{mean\;stepsize}$) was smaller than 2.11 [40]. The PSD mask was created based on the maximum intensity projection of Homer1c-mCherry. Peaks in intensity were detected, after which a FWHM-like boundary was defined for each PSD. 

Transient confinement analysis on mobile trajectories was done in MATAB using slightly modified scripts from a previously published MATLAB implementation [21]. As modifications, we included the last segments of the trajectory shorter than Sm in the averaging to improve confinement detection at the end of a trajectory (Appendix A). Furthermore, the point in the trajectory is the middle of the sliding window of the *D* over time instead of the beginning of the sliding window, for a better correlation with *L* over time (Appendix A). Lastly, we corrected the ln into the log10 for defining confinement index *L*. As *D*, the maximum *D_inst_* was used, estimated for each sub-trajectory of Δ10. Parameters used in the analysis are: *Lc* = 5.5, *Sm* = 15, α = 0.5, *Tc* = 0.2 s (Appendix A). The confinement zones are further analyzed for size and duration of confinement and diffusion coefficient in and outside confinement zones. Confinement maps were created based on the detected confinement radius for each confinement zone. Each confinement zone was stored as a two-dimensional Gaussian distribution with the radius as FWHM. The final matrix was plotted with a color-code, where higher values indicate confinement hotspots because there are multiple Gaussians on top of each other. 

### 2.9. Statistical Analysis

Statistical significance was tested using a Student’s *t*-test when comparing two groups. Statistical tests with a *p*-value below 0.05 were considered significant. Significance is indicated as follows: *p* < 0.05 is indicated by *, *p* < 0.01 by **, and *p* < 0.001 by ***. Analysis was performed on neurons originating from three individual preparations of hippocampal neurons. The number of neurons used for analysis is indicated as n. Statistical analysis and graphs were prepared in GraphPad Prism (version 9, Graphpad Software, San Diego, CA, USA) and figures were generated in Adobe Illustrator 2022 (Adobe Systems, San Jose, CA, USA).

## 3. Results

### 3.1. Detection of Transient Confinement within Single-Molecule Trajectories

To detect transient confinement within single-molecule trajectories, Simson et al. [11] developed a statistical framework defining areas where a diffusing molecule remains significantly longer than expected if it would undergo Brownian motion. The probability ψ that a molecule stays within a bounded region for a set period of time has been defined by Saxton as: (1)$$\log\psi =0.2048-2.5117\left(Dt/R^{2}\right)$$
with *D* being the diffusion coefficient, *t* the period of time, and *R* the radius of the region [20]. Simson et al. [11] used this probability to derive the confinement index *L*, which is inversely related to ψ: (2)$$L=\left\{\begin{array}{cc} -\log\left(\psi \right)-1) & \psi \leq 0.1\\ 0 & \psi >0.1 \end{array}\right.$$

High values of *L* are thus indicative of non-random confinement and *L* = 0 in cases of random diffusion. The calculation of *L* is inherently based on a number of variables that will influence the detection power of the confinement analysis. To correctly interpret the results of this analysis, here we aimed to understand the influence of these different variables on the detection power of the analysis. The analysis in this study is based on the MATLAB implementation from Ménchon et al. [21] with some modifications (see Section 2.8; Appendix A). To define the confinement index over time for individual trajectories, a trajectory is first divided into segments with minimum length *Smin* and maximum length *Sm* (Figure 1A) and for each individual segment, *L* is calculated as described above. As such, these parameters are thus expected to determine the spatial and temporal sensitivity of the analysis. The radius *R* of the segment is defined by the maximum distance between one of the points and the starting point of the segment. The diffusion coefficient *D* used in the formula is the same for all the segments in the trajectory and should represent the diffusion coefficient of the molecule if it would move freely. Then, for each point in the trajectory, *L* is averaged over all the segments that include that point. Note, that as a consequence, the averaged *L* for points at the beginning and the end of the track is averaged over fewer segments than points in the middle. As can be seen for a simulated trajectory with a short period of transient confinement (Figure 1B), *L* values are higher during the confined period, which is accompanied by a temporal decrease in the diffusion coefficient (Figure 1C). Confined periods within a trajectory can then be defined by selecting periods in which all points are above the critical minimum *L* (*Lcm*) for a duration longer than a critical period of time (*Tc*) (Figure 1C).

When increasing the diffusion coefficient *D* of simulated molecules, it can be seen that because for a molecule with a higher *D* it is less likely that it would stay in the same area for the same time, relatively higher *L* values are calculated (Figure 1D and Appendix A). On the other hand, when *D* remains constant but the region in which the molecule is observed is larger, the calculated confinement index becomes lower, because it is more likely that a freely moving molecule would stay in a larger area (Figure 1E and Appendix A). Lastly, if we only increase the observed time window (*t*)—which is represented by *Smin* and *Sm*—while keeping the other variables the same, a higher confinement index will be detected because it is less likely that a Brownian molecule with that diffusion coefficient would stay in the same area for a longer time (Figure 1F).

### 3.2. Optimizing Input Parameters for Accurate Transient Confinement Zone Detection in Single-Molecule Trajectories

After considering the influence of the different variables in the confinement index calculation, we aimed to optimize the user-defined parameters for the confinement detection to minimize the false positive rate while maximizing the true detection rate. False positives can be detected because *L* is not zero: even in random walks fluctuations in *L* could be inadvertently interpreted as confinement. 

First, we wanted to define what effect the input parameters would have on incorrectly detecting confinement in random walks, therefore we simulated random walks with parameters comparable to experimental situations (*D* = 0.05 μm^2^/s, *dt* = 0.02 s, 100 frames) and started off with a maximum window size (*Sm*) of 15 frames. Lowering the minimum *L* threshold (*Lcm*) increased the percentage of false positives detected in random walks, similar to decreasing the critical period of time (*Tc*) (Figure 2A). Changing the maximum window size (*Sm*) did not lead to a large change in the detection of false positives (Figure 2B). Thus, to achieve the lowest number of false positives, *Lcm* and *Tc* should be set relatively high, whereas the value of *Sm* is of less significance. 

Next, we investigated the effect of the input parameters on the detection of true confinement. Therefore, we simulated trajectories with confined periods and parameters comparable to experimental situations (*D* = 0.05 μm^2^/s, *dt* = 0.02 s, 100 frames, confined period of 15 or 50 frames and simulated confinement radius (*Rconf*) of 0.04–0.06 μm). Increasing *Lcm* lowered the detection power of true confinement (Figure 2C). Furthermore, in contrast to the effect on random walks, the time window plays an important role in the correct detection of true confinement. The power of detection increased with larger *Sm* when the confinement period was equally long, whereas the power of detection decreased if the confinement period was shorter than *Sm* (Figure 2D). Choosing this maximum time window will also affect the power of detecting multiple confinement zones after each other. Increasing the *Sm* to achieve increased power of detection could lead to combining multiple confinement zones because *L* will not go below the threshold in between the zones (Figure 2E, orange and green traces). Taken together, to achieve high power of detection, *Lcm* should not be too high and the value for *Sm* will affect the resolution at which confinement zones can be detected. While higher *Sm* values increase detection of confinement, it reduces the power to detect shorter periods of confinement that, at high *Sm* values, will be averaged out or combined with other confinement zones. 

Because the diffusion coefficient can change over the course of a trajectory, Ménchon et al. introduced another *L* threshold where the threshold is defined by the average *L* of a trajectory multiplied by a factor *α* that ranges from 0 to 1 [21]. It is important to consider the effect of using this threshold definition as it can both increase and decrease the power of detection. For example, when the average *L* is high because of two highly confined regions, *Lc* will be higher than *Lcm,* resulting in not detecting a less confined third region (Figure 2F). On the other hand, when there are highly confined zones—possibly caused by a lower diffusion coefficient—shortly after each other, using the threshold *Lc* will separate the two zones, whereas *Lcm* would be too low (Figure 2G). 

### 3.3. Estimated Diffusion Coefficient Can Be Influenced by Track Length

One important parameter in the confinement index formula is the diffusion coefficient. This is the diffusion coefficient of the molecule if it could move freely. However, setting this diffusion coefficient is not trivial. One approach is to use the maximum instantaneous diffusion coefficient (*D_inst_*) per trajectory, assuming that the molecule undergoes Brownian diffusion during at least part of the trajectory [21]. To calculate the maximum *D_inst_* of a trajectory, we need to determine how the diffusion coefficient changes along the trajectory. Therefore, the *D_inst_* was estimated based on the linear part of the MSD versus time lag curve for each window of ten steps over the complete trajectory and the maximum *D_inst_* was used as the *D* of that trajectory. For this short window, the uncertainty in estimating the diffusion coefficient could be relatively high, but it allows for the detection of changes in the diffusion coefficient within individual trajectories. Another approach would be to test against a set diffusion coefficient chosen a priori that would resemble the Brownian diffusion of the molecule of interest. This value could for instance be deduced from independent tracking experiments, taking the diffusion coefficient of a freely moving population of trajectories as a reference value [22,28,29].

We investigated the effect of the different approaches in defining the Brownian diffusion coefficient on the false positive rate and confinement detection. For simulations, it is possible to test against a set diffusion coefficient and we compared this approach (Dset) to using the maximum instantaneous diffusion coefficient (Dmax). An interesting observation was that the false positive rate depended on the length of the simulated tracks for the Dmax approach whereas this was stable with Dset (Figure 3A,B). This was caused by a higher Dmax for the longer tracks, probably because there is a higher chance that there is a peak in the diffusion coefficient over time when there are more steps in the trajectory (Figure 3C). Similarly, in confined tracks with the same confinement zone but a longer total length, the confinement index during the period of confinement was higher with the Dmax method, whereas with Dset, the confinement index was similar for all track lengths (Figure 3D,E). This difference could also be explained by the higher Dmax values for the longer tracks (Figure 3F). To improve the Dmax method, we tested the effect of including more steps in the sliding window for the estimation of *D_inst_*. That resulted in a lower Dmax because the peaks are averaged out. However, Dmax still increases with longer track lengths (Figure 3G). A third option (Dinst) would be to use the estimated instantaneous diffusion coefficient per track as the *D* in the confinement formula as used by Simson et al. [11]. However, although based on only the first three points of the MSD versus time lag curve, Dinst is heavily influenced by confinement in the trajectory, whereas Dmax only declined when the largest part of the trajectory is confined (Figure 3H). Thus, using *D_inst_* per trajectory negatively affects the ability to detect transient confinement. Particularly, when a track was confined for more than 50% of the time, the power of detection dropped dramatically (Figure 3I). To conclude, using a set diffusion coefficient as *D* in the confinement formula appears to be the most reliable method, however this is under the assumption that the Brownian diffusion coefficient of the molecule is known. The alternative method would be using the Dmax, where the diffusion of the whole trajectory is tested against the fastest segment in the trajectory, thus investigating changes in diffusion behavior over the course of the trajectory.

### 3.4. Detection Limits in the Confinement Analysis

Next, we wanted to understand the detection limits of the confinement analysis to interpret the results correctly. Can similarly sized confinement zones still be detected for molecules with different diffusion coefficients? Purely based on the theoretical formula behind the confinement analysis, it is only possible to detect a large confinement zone if the diffusion coefficient is high enough and the time is long enough (Figure 4A,B). Only then will the confinement index *L* still reach the threshold when the confinement radius is large. In other words, the ratio between the possible explored area of a random diffusing molecule (*D* × *t*)—with given diffusion coefficient and time—and the observed explored area (*R^2^*) should be high enough to be considered confined (Figure 4C). To study the limits of the confinement analysis in detecting confinement zones of specific sizes, we simulated confined tracks with varying diffusion coefficients (0.01–1 μm^2^/s) and confinement radii (*Rconf,* 20–200 nm). Next, we ran the transient confinement analysis on these simulations and found the fraction of confinement zones that were detected and their detected confinement radius (Figure 4D,E). It is clear from the results that the larger confinement zones cannot be detected in tracks with low diffusion coefficients. Such a molecule with a low diffusion coefficient could reside in a larger area without deviating from a Brownian molecule in their behavior. 

Additionally, the first and last points in the confinement profile of a track are averaged over a lower number of segments. Therefore, we tested whether the timing of a confinement zone would affect the power of detection. We simulated confinement zones with varying radii and duration of 15 or 50 frames at the beginning, middle, and end of the track, and observed no considerable differences in the detected confined displacements (Figure 4F and Appendix A). Thus, the detection of confinement in this analysis is not limited by the timing of the confinement periods. 

### 3.5. Influence of Localization Error and Frame Rate on Confinement Detection Accuracy

In single-molecule tracking analysis, the first step is to localize the sub-pixel position of fluorescence emission events of single molecules. Generally, the point spread function of single-molecule emission spots can be fitted with a two-dimensional Gaussian function. The uncertainty inherent to the fitting routine, or localization error, influences the accuracy of the detected single-molecule trajectories. To investigate the effect of the localization error on the performance of the confinement detection, we simulated random walks and confined tracks with no error, 20 nm, and 50 nm localization error. We found that in random walks, a larger localization error resulted in fewer false positives (Figure 5A and Appendix A). Thus, the *Lcm* threshold could be lowered to achieve the same detection precision. This can be explained by *R^2^* being, on average, larger with a higher localization error. However, a higher localization error also resulted in a dramatic decrease in the detection of both short and longer confinement periods, even with the lowered Lcm threshold (Figure 5B,C and Appendix A).

Another experimental parameter we considered is the frame rate of the acquisition that determines the time resolution in the single-molecule trajectories. It is important to adjust the parameters of the analysis accordingly, especially when comparing different experiments with different experimental parameters. Using the same maximum segment window with a smaller interval time will lead to narrower peaks in random walks (Figure 5D,E). Moreover, for a similar long period of confinement, the average confinement index will be lower with a smaller interval time and the same segment window (Figure 5G,H). Changing the interval time can be corrected by adjusting the maximum segment window (*Sm*) accordingly to obtain a more similar confinement index over time profile (Figure 5F,I,J).

### 3.6. Spatial Mapping of Transient Confinement of Membrane Probes

Next, we wanted to test the robustness of the confinement analysis on experimental single-molecule tracking data. We therefore focused on the neuronal membrane, where the dynamic behavior of individual membrane components collectively contributes to the efficient transfer of synaptic signals [41]. Subsynaptic domains enriched in glutamate receptors and scaffolding molecules were found to be aligned with the presynaptic glutamate release to optimize synaptic transmission, which emphasizes the importance of the heterogeneous organization of the neuronal membrane [42,43]. First, we developed a visualization tool to map the detected confinement zones resulting in a spatial heatmap of confinement hotspots. Such a heatmap can be displayed together with other cellular markers to locate confinement hotspots relative to specific subcellular domains. To create such a heatmap, we plotted every confinement zone as a Gaussian with amplitude 1 and the radius of the confinement zone as the FWHM (Figure 6A). Overlapping confinement zones would result in higher amplitude values, resulting in a heatmap that is color-coded for the amplitude values. Neurons were transfected with GFP-GPI, and we performed single-molecule tracking experiments using an anti-GFP nanobody coupled to Atto647N (Figure 6B). After tracking the molecules, we performed the transient confinement analysis on the mobile trajectories to obtain information on the location and timing of confinement zones in experimental trajectories (Figure 6C,D). Finally, we plotted the trajectories and confinement heatmap in combination with a marker of excitatory synapses (Homer1c-mCherry) (Figure 6E–J). We observed confinement of GFP-GPI in the axon, dendrite, and spines, where confinement does not seem to be specifically enriched at synaptic locations. Overall, this visualization tool allows for the detection of areas with high levels of transient confinement in a cellular context, by mapping the confinement zones relative to cellular markers. 

To address the effect of changing parameters in the confinement analysis on the resulting confinement measures in experimental datasets, we applied the confinement analysis on a single-molecule tracking acquisition of GFP-GPI in the neuronal membrane. Lowering the *Lcm* increased the detected confinement radius, the diffusion coefficient inside confinement zones, and resulted in more trajectories with detected confinement (number of tracks with confinement for *Lcm* 3: 381, *Lcm* 4: 309, *Lcm* 5.5: 219, Figure 7A and Appendix A). On the other hand, varying *Tc* did not affect the average detected confinement radius or the diffusion coefficient inside confinement zones, although with higher *Tc*, fewer trajectories with detected confinement were found because of the stricter thresholds (*Tc* 0.1: 311, *Tc* 0.2: 219, *Tc* 0.3: 139, Figure 7B and Appendix A). Lastly, a longer time window resulted in the detection of larger confinement zones and a higher diffusion coefficient inside confinement zones, but did not noticeably affect the number of trajectories with detected confinement (*Sm* 5: 181, *Sm* 15: 219, *Sm* 30: 177, Figure 7C and Appendix A).

Furthermore, we wanted to investigate the possibility to compare confinement of different membrane probes with different diffusional properties. Therefore, we compared the confinement behavior of (1) the AMPAR subunit GluA2 with (2) an artificial protein with single-spanning transmembrane domain (GT46), (3) GFP-GPI, and (4) the phospholipid DOPE in neurons. We detected transient confinement in the trajectories of all four membrane probes, but considerably larger confinement zones were detected for DOPE in comparison to GluA2 (Figure 7D). This difference could be explained by their difference in diffusion as DOPE has a higher diffusion coefficient (Figure 4, Figure 7E,F, and Appendix A). The difference in size of the confinement zones related to their difference in diffusion coefficient for all probes, except GPI which diffused slightly faster than GT46 but revealed slightly smaller confinement zones. In conclusion, comparison of confinement zone properties should be approached with caution when the diffusion coefficients of the probes differ considerably.

Lastly, we aimed to compare the transient confinement properties of two glutamate receptors that have comparable diffusion coefficients: the metabotropic glutamate receptor 5 (mGluR5) and the AMPA receptor subunit GluA2 (Appendix A). These receptors both reside in the postsynaptic membrane, but GluA2 is found to be more enriched in the synapse whereas mGluR5 is concentrated in an area around the synapse [44,45]. We found that the diffusion coefficient of GluA2 inside confinement zones was lower than for mGluR5 (mGluR5: 0.00891 ± 0.00035 μm^2^/s and GluA2: 0.00793 ± 0.00025 μm^2^/s; Figure 7G), while the diffusion coefficient outside confinement zones was not significantly different (mGluR5: 0.0813 ± 0.0030 μm^2^/s and GluA2: 0.0789 ± 0.0040 μm^2^/s; Figure 7H). In addition, the time GluA2 resided in confinement zones was longer than mGluR5 (mGluR5: 0.419 ± 0.012 s and GluA2: 0.477 ± 0.010 s; Figure 7I), while the total track length was not significantly different (mGluR5: 1.65 ± 0.043 s and GluA2: 1.71 ± 0.054 s; Figure 7J). These results suggest that there are different mechanisms responsible for the confinement of mGluR5 and GluA2.

## 4. Discussion

Single-molecule tracking is a powerful approach to investigate the heterogeneous organization of cellular membranes. However, precise and reliable analysis of single-molecule tracking data remains challenging. This study set out to gain a better understanding of the detection of transient confinement in single-molecule trajectories.

We found that a considerable factor in the confinement analysis is the diffusion coefficient that is used in the formula to define the diffusion coefficient of the molecule when it would move freely. However, defining this free diffusion coefficient a priori is not straightforward. We therefore compared three different methods: using the maximum instantaneous diffusion coefficient per trajectory based on shorter segments [21], the instantaneous diffusion coefficient per trajectory [11], and defining a constant diffusion coefficient for the molecule of interest considered to be the free diffusion coefficient [22,25,26]. It is important to note the limitations of all these methods and their consequences on the results. The Dmax could be an overestimation of the free diffusion coefficient and therefore lead to an overestimation of the detected confinement in that trajectory and in addition, the value of Dmax seems to depend on the length of the track. However, it does allow for detecting changes in diffusion behavior over the trajectory. Estimating the instantaneous diffusion coefficient per trajectory could be heavily influenced by the percentage of confinement in the trajectory and therefore lead to reduced power of detection. Using a set diffusion coefficient is another approach of using the confinement analysis as with this method not only transient confinement would be detected but also completely confined tracks, as all the tracks are tested against the same diffusion coefficient and thus no longer for a temporal deviation within the trajectory. This heavily depends on choosing the free diffusion coefficient in a correct way. Previous studies used this approach, but this was under the assumption that the diffusion of these molecules was mainly Brownian [22,25,26]. This should be done with caution as we showed that the percentage of confinement in a track could influence the Dinst. It is important to note that when using this approach in the case of comparing confinement after, for instance, an experimental treatment, that the difference may solely come from a change in the diffusion coefficient and does not perse reflect a true difference in the confinement behavior. 

One of the interesting quantitative outcomes of the confinement analysis is the size of a confinement zone. We found, however, that the maximal size that can be detected, depends on the diffusion coefficient of the molecule. Thus, comparing confinement of molecules with varying diffusion coefficients could lead to erroneous conclusions about differences in their confinement zone sizes. It is also not directly evident from the data which biological mechanism underlies these confined periods, and different mechanisms might also result in different confinement measures. Phase separation into liquid-ordered and liquid-disordered phases in model membranes arises mainly from the combination of lipid composition and temperature. Biological membranes are even more complex, both in the heterogeneity of lipids and involvement of membrane-associated proteins that affect the thermodynamics. Neurons and more specifically synapses are enriched in cholesterol and sphingolipids [46,47]. These lipids comprise the more ordered phase in the membrane and especially add up to the heterogeneous organization of biological membranes [48,49,50]. Furthermore, characteristics of confinement zones could be explained by compartmentalization by the actin-based membrane skeleton, molecular crowding, or binding to scaffold proteins [37,38,39]. 

Interestingly, we detected differences in the confinement behavior of glutamate receptors mGluR5 and GluA2, which suggests that different mechanisms are responsible for their confinement. It is known that these glutamate receptors show different distribution patterns within the postsynaptic membrane, GluA2 is concentrated in the postsynaptic density (PSD) whereas mGluR5 seems to be surrounding the PSD [51]. As our results reveal that GluA2 is trapped for longer time and moves slower, this could be an indication that GluA2 is trapped at a specific location mainly by scaffold molecules, supported by the finding that the actin cytoskeleton is absent from the PSD [52]. On the other hand, mGluR5 might be slowed down in its diffusion by other mechanisms such as steric hindrance or interactions with the cytoskeleton. Other methods for confinement analysis might help in detecting specific underlying mechanisms. For example, Meilhac et al. modified the method from Simson et al. to be able to detect jumps between compartments in the membrane [11,12]. Another way to characterize confinement are potential wells, describing a field of force resulting from molecular interactions, which has been suggested for synaptic receptors [18,40,41]. Lastly, this method could also be applied to less complex membranes to get a better estimate of the mechanism behind detected confinement behavior. For example, to study the effect of temperature on confined behavior and thus membrane organization.

## 5. Conclusions

This study provides a systematic analysis of the influence of the different parameters that are used for detecting temporal confinement in single-molecule trajectories and proposes a visualization tool to map confinement zones in the cellular context. To demonstrate the validity of the presented approach, we studied the diffusion of two glutamate receptor types in neurons and found that mGluR5 and GluA2 differ in their confinement behavior. We believe the presented results can guide future studies in the correct detection and interpretation of confinement analysis. As such, this study will help in better understanding the complex organization of the membrane by reliably detecting and spatially mapping confinement.

## Figures and Tables

**Figure 1 membranes-12-00650-f001:**
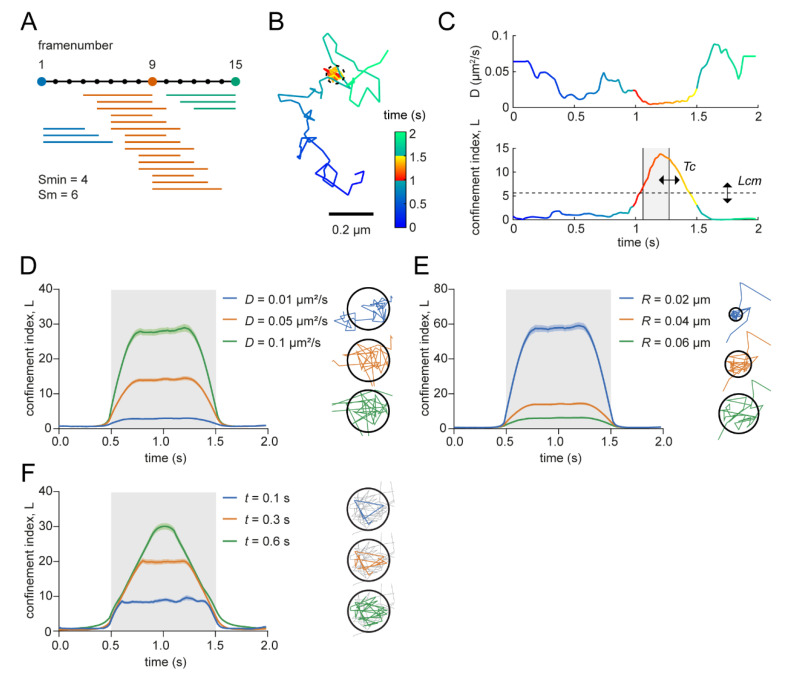
Influence of variables in confinement index formula. (**A**) Schematic diagram of how trajectories are divided into segments with different lengths. In this simplified example, the minimal segment length (*Smin*) is 4 frames and the maximum segment (*Sm*) is 6. All the segments are displayed that will be included in the confinement index of frame 1 (blue), 9 (orange), and 15 (green). (**B**) Simulated trajectory with a confinement zone with radius of 40 nm. Color-coded for time and confined period. Scale bar, 0.2 μm. (**C**) Diffusion coefficient (top) and confinement index (bottom) over time of the simulated trajectory in (**B**). Color-coded for time and confined period. *Tc* and *Lcm* indicate the critical time period and critical *L* threshold used to detect confinement periods. (**D**) Average confinement index over time for 100 simulated trajectories with three different diffusion coefficients. Simulated confined period is indicated with the gray box. Illustrations display for each condition one example segment of the track in the corresponding color. Simulated confinement radius (*Rconf*) is 0.04 μm and *Sm* is 15 frames. (**E**) Average confinement index over time for 100 simulated trajectories with three different simulated confinement radii. Simulated confined period is indicated with the gray box. Illustrations display for each condition one example segment of the track in the corresponding color. Diffusion coefficient is 0.05 μm^2^/s and *Sm* is 15 frames. (**F**) Average confinement index over time for 100 simulated trajectories with 3 different segment lengths (*Sm* = *Smin*). Simulated confined period is indicated with the gray box. Illustrations display for each condition one example segment of the track in the corresponding color. Diffusion coefficient is 0.05 μm^2^/s and *Rconf* is 0.04 μm. Data are represented as means ± SEM.

**Figure 2 membranes-12-00650-f002:**
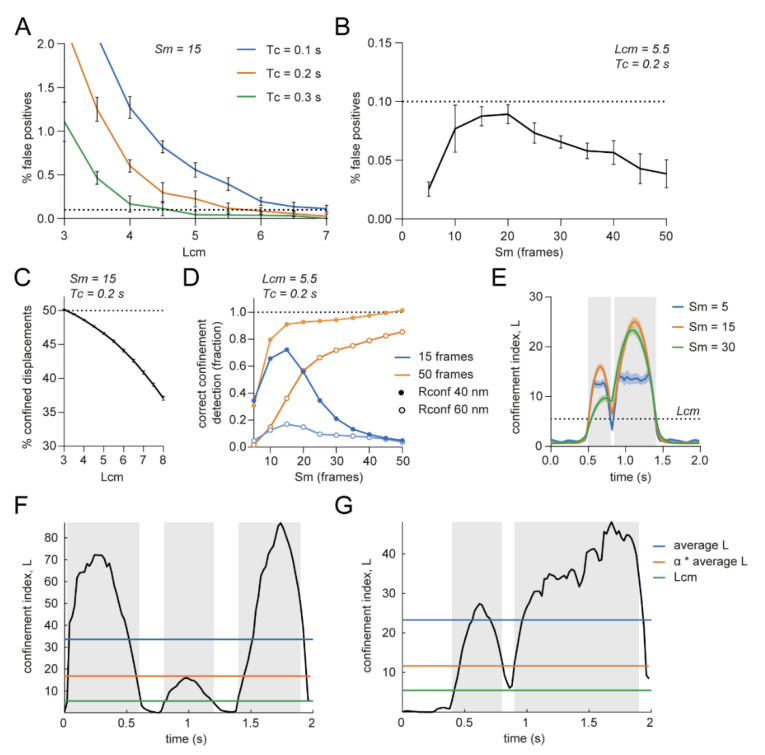
Influence of user-defined parameters on the confinement analysis. (**A**) Effect of minimal critical *L* (*Lcm*) and critical time (*Tc*) on percentage of false positives in simulated random walks. Dotted line indicates 0.1% false positives. *Sm* is 15. Five independent simulations of 100 trajectories. Data are represented as means ± SEM. (**B**) Effect of *Sm* on percentage of false positives in simulated random walks. Dotted line indicates 0.1% false positives. *Lcm* is 5.5 and *Tc* is 0.2 s. Five independent simulations of 1000 trajectories. Data are represented as means ± SEM. (**C**) Effect of *Lcm* on the percentage of detected confined displacements in trajectories simulated to be confined for 50 of the 100 frames. *Sm* = 15, *Tc* = 0.2 s, *Rconf* = 0.04 μm. Five independent simulations of 1000 trajectories. Data are represented as means ± SD. (**D**) Effect of *Sm* on correct confinement detection in trajectories simulated to be confined 15 or 50 of the 100 frames. *Lcm* = 5.5 and *Tc* = 0.2 s. One hundred simulated trajectories per condition. (**E**) Effect of *Sm* on the ability to discriminate between multiple confinement zones shortly after each other. *Lcm* = 5.5 and *Rconf* = 0.03 μm, 100 trajectories. Simulated confined periods are indicated with the gray boxes. Data are represented as means ± SEM. (**F**) Negative effect of ‘*α* * average *L*’ as *Lc*. Single simulated trajectory with three confinement zones. The middle confinement zone does not reach the threshold. *Rconf* = 0.015 μm and 0.03 μm and *α* = 0.5. Simulated confined periods are indicated with the gray boxes. (**G**) Positive effect of ‘*α* * average *L*’ as *Lc*. Single simulated trajectory with two confinement zones. With *Lcm* as threshold, both confinement zones would have been merged into one. *Rconf* = 0.03 μm and *α* = 0.5. Simulated confined periods are indicated with the gray boxes.

**Figure 3 membranes-12-00650-f003:**
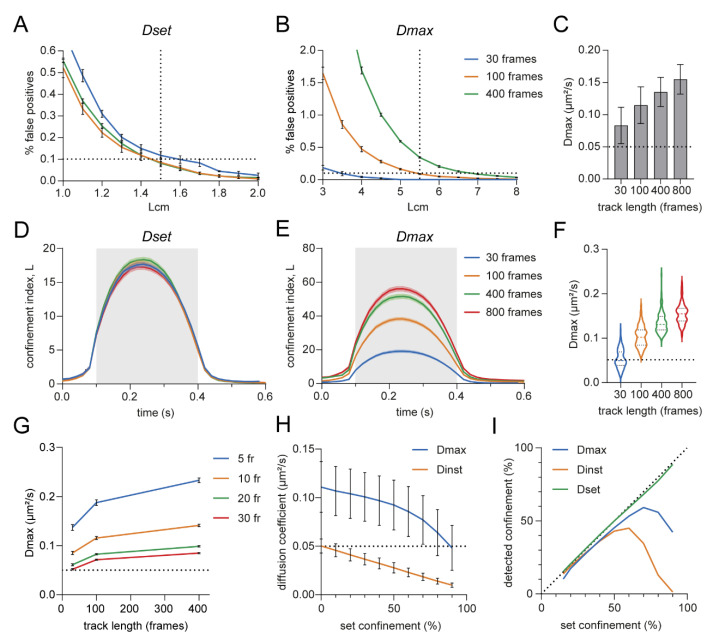
Effect of diffusion coefficient in the confinement index formula on confinement detection. (**A**,**B**) Effect of track length on the percentage of false positives in random walks using Dset (**A**) or Dmax (**B**) as diffusion coefficient in the confinement index formula. Five independent simulations of 1000 trajectories per condition. Data are represented as means ± SEM. (**C**) Effect of track length on Dmax. One hundred simulated random walks per condition. Data are represented as means ± SD. (**D**,**E**) Effect of track length on the confinement index over time using Dset (**D**) or Dmax (**E**) as diffusion coefficient in the confinement index formula. 100 confined trajectories per condition, *Rconf* = 0.02 μm. Data are represented as means ± SEM. (**F**) Violin plots of Dmax values from (**E**). (**G**) Effect of the length of the segments used for estimating *D_inst_* within trajectories on Dmax. (**H**) Effect of the percentage of confinement within a trajectory on the estimated diffusion coefficient of the whole trajectory (Dinst) or the Dmax. One thousand trajectories per condition, *Rconf* = 0.04 μm. Data are represented as means ± SD. (**I**) Effect of the percentage of confinement within a trajectory on the detected confinement within the trajectory using Dmax, Dinst or Dset. Dotted line indicates set confinement = detected confinement. One thousand trajectories per condition, *Rconf* = 0.04 μm.

**Figure 4 membranes-12-00650-f004:**
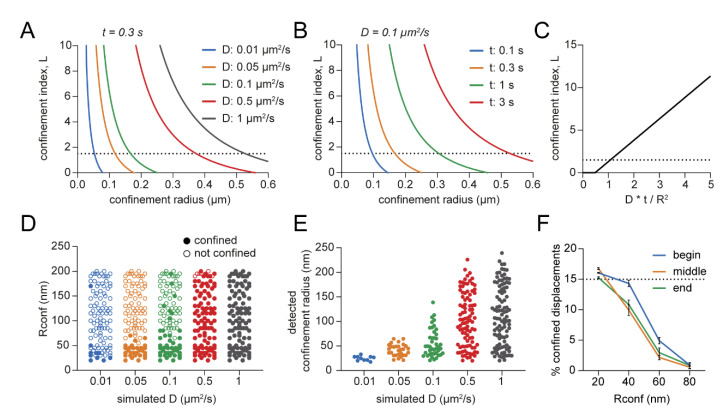
Detection limits of the confinement analysis. (**A**) Theoretical effect of the confinement radius and diffusion coefficient on the confinement index that can be detected. Observing time = 16 frames or 0.3 s. Dotted line indicates *Lcm* of 1.5 as used with Dset. (**B**) Theoretical effect of the confinement radius and observing time on the confinement index that can be detected. *D* = 0.1 μm^2^/s. (**C**) Theoretical relation between *D × t/R^2^* and the confinement index. (**D**) Effect of the diffusion coefficient in trajectories on the ability to detect a range of different simulated confinement radii (*Rconf*). (**E**) Effect of the diffusion coefficient in trajectories on the detected confinement radius given the same range of confinement radii in (**D**) as input for the simulations. (**F**) Effect of the timing of simulated confinement periods and *Rconf* on the correct detection of confinement. Fifteen frames confined of total 100 frames per simulated track. Five independent simulations of 100 trajectories per condition. Dotted line indicates correct confinement detection. Data are represented as means ± SD.

**Figure 5 membranes-12-00650-f005:**
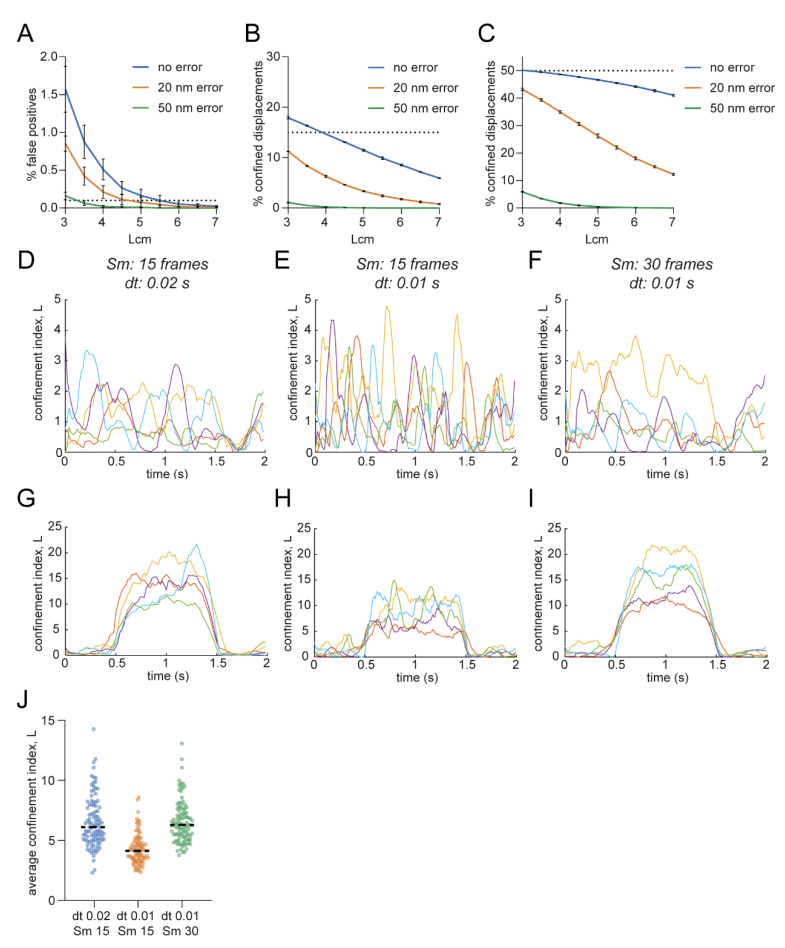
Influence of experimental parameters on confinement detection. (**A**) Effect of localization error on the percentage of false positives detected in random walks. Three independent simulations of 1000 trajectories per condition. Data are represented as means ± SD. (**B**,**C**) Effect of localization error on the percentage of detected confined displacements in tracks simulated to be confined for 15 (**B**) or 50 (**C**) of the 100 frames. *Rconf* = 0.04 μm. Three independent simulations of 1000 trajectories. Data are represented as means ± SD. (**D**–**F**) Confinement index of 5 random trajectories over time with (**D**) *Sm* = 15, *dt* = 0.02 s and 100 frames, (**E**) *Sm* = 15, *dt* = 0.01 s and 200 frames, (**F**) *Sm* = 30, *dt* = 0.01 s and 200 frames. (**G**–**I**) Confinement index of 5 confined trajectories over time with (**G**) *Sm* = 15, *dt* = 0.02 s and 100 frames, (**H**) *Sm* = 15, *dt* = 0.01 s and 200 frames, (**I**) *Sm* = 30, *dt* = 0.01 s and 200 frames. *Rconf* = 0.04 μm. (**J**) Corresponding average confinement index values of (**G**–**I**).

**Figure 6 membranes-12-00650-f006:**
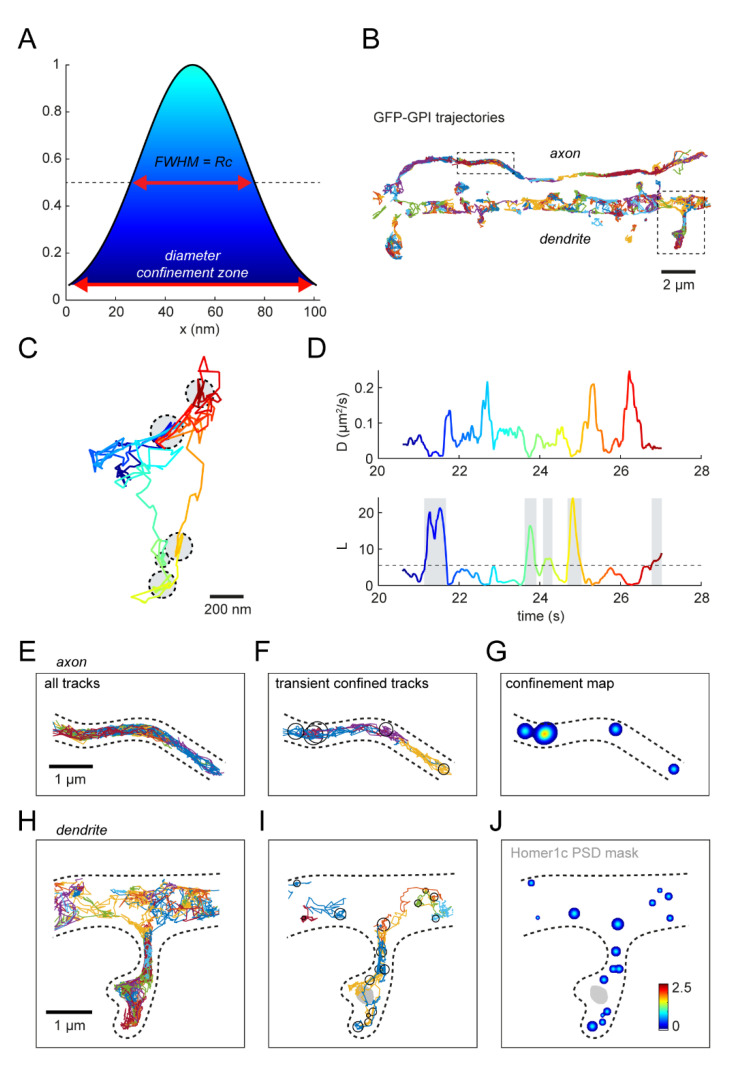
Spatial visualization of confinement. (**A**) Gaussian function used in confinement map with amplitude 1. FWHM is the confinement radius. Color-coded for amplitude. (**B**) GFP-GPI anti-GFP nanobody Atto647N trajectories in axon and dendrite. Boxes mark location of zooms in (**E**–**J**). Scale bar, 2 μm. (**C**) Example experimental GPI trajectory with 5 confinement zones indicated with the grey circles. Color-coded for time. Scale bar, 200 nm. (**D**) Diffusion coefficient (top) and confinement index (bottom) over time of the experimental trajectory in (**C**). Color-coded for time and the detected confined periods are indicated with the gray boxes. (**E**–**J**) Zoom of trajectories in the axon (**E**–**G**) and dendrite (**H**–**J**). (**E**,**H**) All trajectories displayed in different colors. (**F**,**I**) Trajectories with confined periods. Confinement zones are indicated with the black circles. (**G**,**J**) Confinement map indicating areas with multiple confinement zones close to or on top of each other. Homer1c PSD mask is indicated in gray in the dendrite. Scalebar, 1 μm.

**Figure 7 membranes-12-00650-f007:**
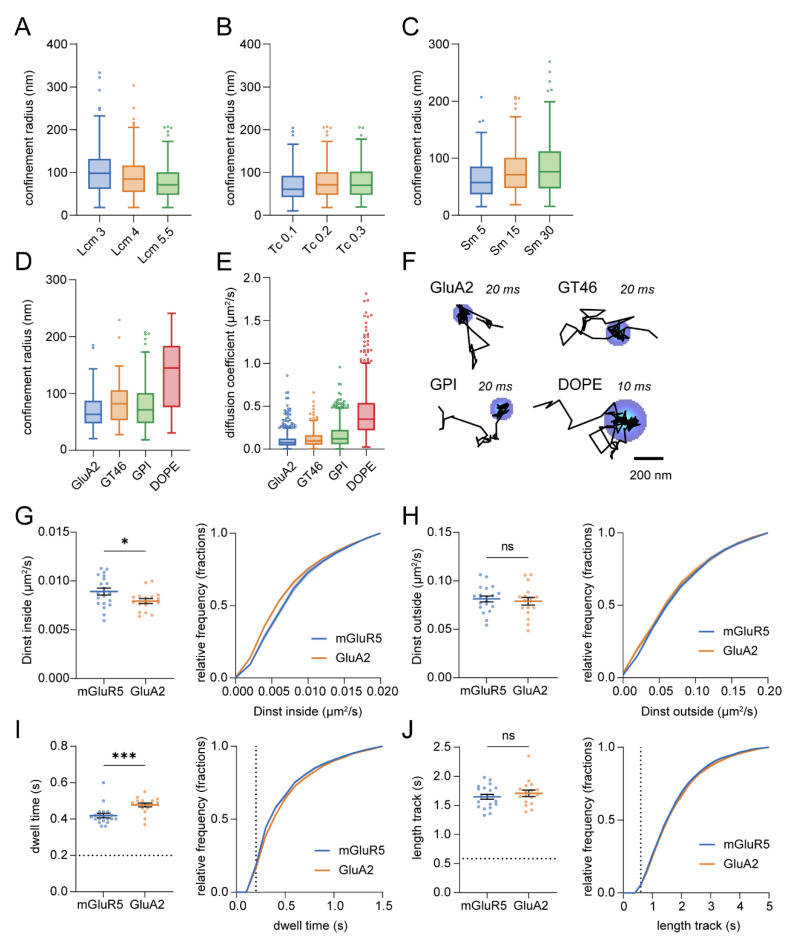
Confinement analysis on experimental trajectories. (**A**–**C**) Effect of user-defined parameters on the detected confinement radius. (**A**) Vary critical *L*, *Lcm*. Number of trajectories with confinement: *Lcm* 3: 381, *Lcm* 4: 309, *Lcm* 5.5: 219. (**B**) Vary critical time, *Tc*. Number of trajectories with confinement: *Tc* 0.1: 311, *Tc* 0.2: 219, *Tc* 0.3: 139. (**C**) Vary maximum segment length, *Sm*. Number of trajectories with confinement: *Sm* 5: 181, *Sm* 15: 219, *Sm* 30: 177. (**D**–**F**) Confinement zones detected for membrane probes: GluA1, GT46, GPI, and DOPE. (**D**) Detected confinement radius for each probe. (**E**) Estimated diffusion coefficients for each probe. (**F**) Example trajectories with their detected confinement zone. Interval times during the acquisition are indicated. Scale bar, 200 nm. (**G**–**J**) (**G**) Average estimated diffusion coefficient inside confinement zones, (**H**) average estimated diffusion coefficient outside confinement zones, (**I**) average dwell time, (**J**) average total track length for mGluR5 (*n* = 20) and GluA2 (*n* = 17; unpaired *t*-test) (left), and corresponding cumulative frequency distribution (right). (**I**) Dotted line indicates dwell time threshold. (**J**) Dotted line indicates minimum track length threshold. Data are represented as means ± SEM. * *p* < 0.05, *** *p* < 0.001, ns, *p* > 0.05.

## Data Availability

The data that support the findings of this study are available on request from the corresponding author, H.D.M.

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
