# Peer review of "Precise Detection and Visualization of Nanoscale Temporal Confinement in Single-Molecule Tracking Analysis"

_membranes, 2022, doi:10.3390/membranes12070650_

Round 1
Reviewer 1 Report
In this work, Manon Westra and Harold D. MacGillavry try to understand the influence of individual parameters on the performance of the confinement analysis and tested the robustness of this analysis on simulated random walks and trajectories that display transient confined behavior. This strategy allowed them to optimize parameters and balance the detection of false-negatives and false-positives. Moreover, a tool to visualize confinement areas in heatmaps was developed for spatial mapping of confinement hotspots relative to subcellular markers. Finally, the analysis on experimental data for a variety of membrane components was tested and two neuronal glutamate receptors, mGluR5 and AMPA-type glutamate receptors with different confinement properties were revealed. Notably, the formation of nanoscale domains in the membrane is of particular interest as these domains underlie critical cellular functions and Single-molecule tracking is a powerful method to detect and quantify molecular motion at high temporal and spatial resolution and the heterogeneous organization of cellular membranes. Therefore, it has been instrumental in understanding mechanisms that underlie membrane organization. However, precise and reliable analysis of single-molecule tracking data remains challenging.This study provide a new method which does not depend on a variety of parameters that heavily depend on experimental factors and thus help in further understanding the dynamic behavior of membrane components and their role in membrane organization. The paper is well written, and many aspects of the new method are tested systematically. There is no doubt that it merits for publication in Membrane. While this study is interesting, the following concerns need to be addressed before publication.
1. More raw simulated trajectories for each condition in Figure 1 should be presented in the supplementary data. Right now there is only one trajectory for each condition.
2.There are several parameters and their corresponding results. A table summarizing them is necessary for better comparison.
3. In this work, the authors tested their model using neuronal membrane, where the dynamic behavior of individual membrane components collectively contributes to the efficient transfer of synaptic signals. It is no double this is an important system. But the author should explain why it is chosen to test. And is this model suitable for other membrane system ? more complex or simple ?
4. A comparison between the result using this new method and the old method can be helpful.
Reviewer 2 Report
The research article titled "Precise detection and visualization of nanoscale temporal confinement in single-molecule tracking analysis" by M. Westra and H.D. MacGillavry presents a detailed analysis to detect temporal confinement in single-molecule trajectories and it values the detection power of such an analysis determining the optimal values for individual parameters. The robustness of the method has been validated on simulated random walks and trajectories that display transient confinement behavior. A heatmap which allows the spatial visualization of confinement has been also developed, together with an experimental validation on two neuronal glutamate receptor, mGluR5 and AMPA-type glutamate receptor.
The manuscript is clear, well-written and well-organized. Both simulations and experimental procedures are systematic; figures are exhaustive and clear. The results are interesting with a high potential of disruptive applications. Discussion is coherent and considers as well recent literature. However, some points need to be addressed before considering it suitable for publication from my side. Recommendations to improve the quality of the manuscript are listed below:
- As a general consideration, I would have expected comments on the role of the temperature. Together with lipid composition it’s one of the main parameters responsible of lipid phase-separation which strongly affect mechanical properties of the membrane including the diffusion within it. What is the temperature in the simulations? For completeness, a mention of such an aspect could be added to frame the issue from a broader point of view.
- In Discussion section, lines 609-11: Authors state “this could be an indication that GluA2 is trapped at a specific location by scaffold molecules, whereas mGluR5 might be slowed down in its diffusion by a mechanism as steric hindrance” In the present form it sounds mainly as a speculative sentence. Another possible explanation could be related to the interaction between GluA2 and cytoskeleton molecules? If not, why? You may add eventually comments on the GluA2 mechanisms already known.
- With reference to experimental measurements, being these performed on neural cells, I would suggest as well a brief overview on the complexity of neural membranes; the authors could refer to the presence of sphingolipids, in particular sphingomyelins, mainly responsible for the extreme complexity in the composition of the neural membrane (the following as well similar references should be added: Biophysical Journal, 101, 2011, 837 https://doi.org/10.1016/j.bpj.2011.07.014; Biophysical Journal, 116,3, 2019, 503 https://doi.org/10.1016/j.bpj.2018.12.018; Journal of Lipid Research, 48, 2, 417, 2007 https://doi.org/10.1194/jlr.M600344-JLR200;) this can help the reader by giving an idea of the complexity of the membrane; at the same time, as a future step, measurements on less complex model systems could be proposed to evaluate whether this type of analysis is able to provide more qualitative information (for example, if it would be able to associate different confinement behaviors to different interaction with membrane components)
- I recommend adding a concluding section; some considerations present in the current version should be moved from the discussion section to the conclusions section. It may be useful for future readers to focus on the key points of the article - the lack of a conclusions section is more suited to a letter or communication than to a research article.
Round 2
Reviewer 2 Report
Authors addressed all the points raised-up in my previous round of revision.